EMBO
reports

# A highly conserved pocket on PP2A-B56 is required for hSgo1 binding and cohesion protection during mitosis

Yumi Ueki[1] (iD), Michael A Hadders[2] (iD), Melanie B Weisser[1] (iD), Isha Nasa[3] (iD), Paula Sotelo-Parrilla[4], Lauren E Cressey[3], Tanmay Gupta[4] (iD), Emil P T Hertz[1], Thomas Kruse[1] (iD), Guillermo Montoya[1] (iD), A Arockia Jeyaprakash[4] (iD), Arminja Kettenbach[3] (iD), Susanne M A Lens[2] (iD) & Jakob Nilsson[1,*] (iD)

## Abstract

The shugoshin proteins are universal protectors of centromeric cohesin during mitosis and meiosis. The binding of human hSgo1 to the PP2A-B56 phosphatase through a coiled-coil (CC) region mediates cohesion protection during mitosis. Here we undertook a structure function analysis of the PP2A-B56-hSgo1 complex, revealing unanticipated aspects of complex formation and function. We establish that a highly conserved pocket on the B56 regulatory subunit is required for hSgo1 binding and cohesion protection during mitosis in human somatic cells. Consistent with this, we show that hSgo1 blocks the binding of PP2A-B56 substrates containing a canonical B56 binding motif. We find that PP2A-B56 bound to hSgo1 dephosphorylates Cdk1 sites on hSgo1 itself to modulate cohesin interactions. Collectively our work provides important insight into cohesion protection during mitosis.

**Keywords** cohesin; mitosis; phosphatase; PP2A-B56; shugoshin
**Subject Categories** Cell Cycle; Structural Biology

## Introduction

The shugoshin proteins (hSgo1 (Sgol1) and hSgo2 (Sgol2) in humans) are conserved protectors of centromeric cohesion by preventing premature release of the cohesin complex (Marston, 2015). The first shugoshin protein was discovered in *Drosophila melanogaster* through the isolation of a mutant, MEI-S332, that lost cohesion prematurely during meiosis (Kerrebrock *et al*, 1992; Kerrebrock *et al*, 1995; Tang *et al*, 1998). Subsequent genetic screens identified the shugoshin proteins in yeast (Katis *et al*, 2004; Kitajima *et al*, 2004; Marston *et al*, 2004). Common to these proteins is the

presence of an N-terminal coiled coil (CC) region that binds to B56 regulatory subunits, hereby localizing PP2A-B56 to centromeres (Kitajima *et al*, 2006; Riedel *et al*, 2006; Tang *et al*, 2006; Xu *et al*, 2009). The function of the yeast PP2A-B56-Sgo1 complex during meiosis is to dephosphorylate Rec8, hereby preventing Separase cleavage of cohesin (Brar *et al*, 2006; Kitajima *et al*, 2006; Riedel *et al*, 2006; Ishiguro *et al*, 2010; Katis *et al*, 2010).

The PP2A-B56 protein phosphatase is a Ser/Thr phosphatase that dephosphorylates numerous substrates to regulate mitosis (Nilsson, 2019; Garvanska & Nilsson, 2020). PP2A-B56 is a trimeric holoenzyme composed of a scaffold subunit (PP2A-A) that connects the B56 subunit with the catalytic subunit (PP2A-C) (Fig 1A) (Xu *et al*, 2006; Cho & Xu, 2007). The B56 subunit of the holoenzyme confers substrate specificity by binding to interactors that target the phosphatase to its substrates. Most B56 interactors bind via a conserved LxxIxE peptide motif that engages a highly conserved pocket on B56 present in all five B56 isoforms (Hertz *et al*, 2016; Wang *et al*, 2016; Wu *et al*, 2017; Wang *et al*, 2020). A number of important mitotic regulators such as BubR1, Kif4A, and RacGAP1 bind to PP2A-B56 through a LxxIxE motif to regulate specific dephosphorylation events. There are five isoforms of B56 (B56α, β, γ, δ, and ε) that display distinct localization patterns during mitosis (Foley *et al*, 2011; Bastos *et al*, 2014; Vallardi *et al*, 2019).

In human somatic cells, hSgo1 recruit PP2A-B56α/ε and to a lesser extent the other PP2A-B56 isoforms, to the centromere (Meppelink *et al*, 2015; Vallardi *et al*, 2019). This protects cohesin complexes from the mitotic prophase pathway by locally antagonizing mitotic kinase activity and thus WAPL mediated removal of cohesin (Salic *et al*, 2004; Kitajima *et al*, 2005; McGuinness *et al*, 2005). Although hSgo2 has been reported to recruit the bulk of PP2A-B56α to centromeres, hSgo2 is not needed for cohesion protection during mitosis (Kitajima *et al*, 2006; Tang *et al*, 2006; Orth *et al*, 2011; Vallardi *et al*, 2019). Instead, hSgo2 protects Rec8 from cleavage by separase during meiosis (Lee *et al*, 2008). In contrast, depleting hSgo1 prevents cohesion protection despite having limited

1   The Novo Nordisk Foundation Center for Protein Research, Faculty of Health and Medical Science, University of Copenhagen, Copenhagen, Denmark
2   Oncode Institute and Center for Molecular Medicine, University Medical Center Utrech, Utrecht University, Utrecht, The Netherlands
3   Biochemistry and Cell Biology, Geisel School of Medicine at Dartmouth College, Hanover, NH, USA
4   Wellcome Center for Cell Biology, University of Edinburgh, Edinburgh, UK
    *Corresponding author. Tel: +45 21328025; E-mail: jakob.nilsson@cpr.ku.dk

effect on PP2A-B56 centromeric levels (Kitajima *et al,* 2006; Tang *et al,* 2006; Vallardi *et al,* 2019). hSgo1 performs cohesion protection through a conserved cohesin binding motif that is absent from hSgo2 (Liu *et al,* 2013; Nishiyama *et al,* 2013; Hara *et al,* 2014). This

hSgo1 cohesin binding motif is phosphorylated by Cdk1 during mitosis on Thr346 to promote cohesin binding (Liu *et al,* 2013). hSgo1 furthermore competes directly with the cohesin release factor WAPL for cohesin binding to prevent WAPL activity (Hara *et al,*

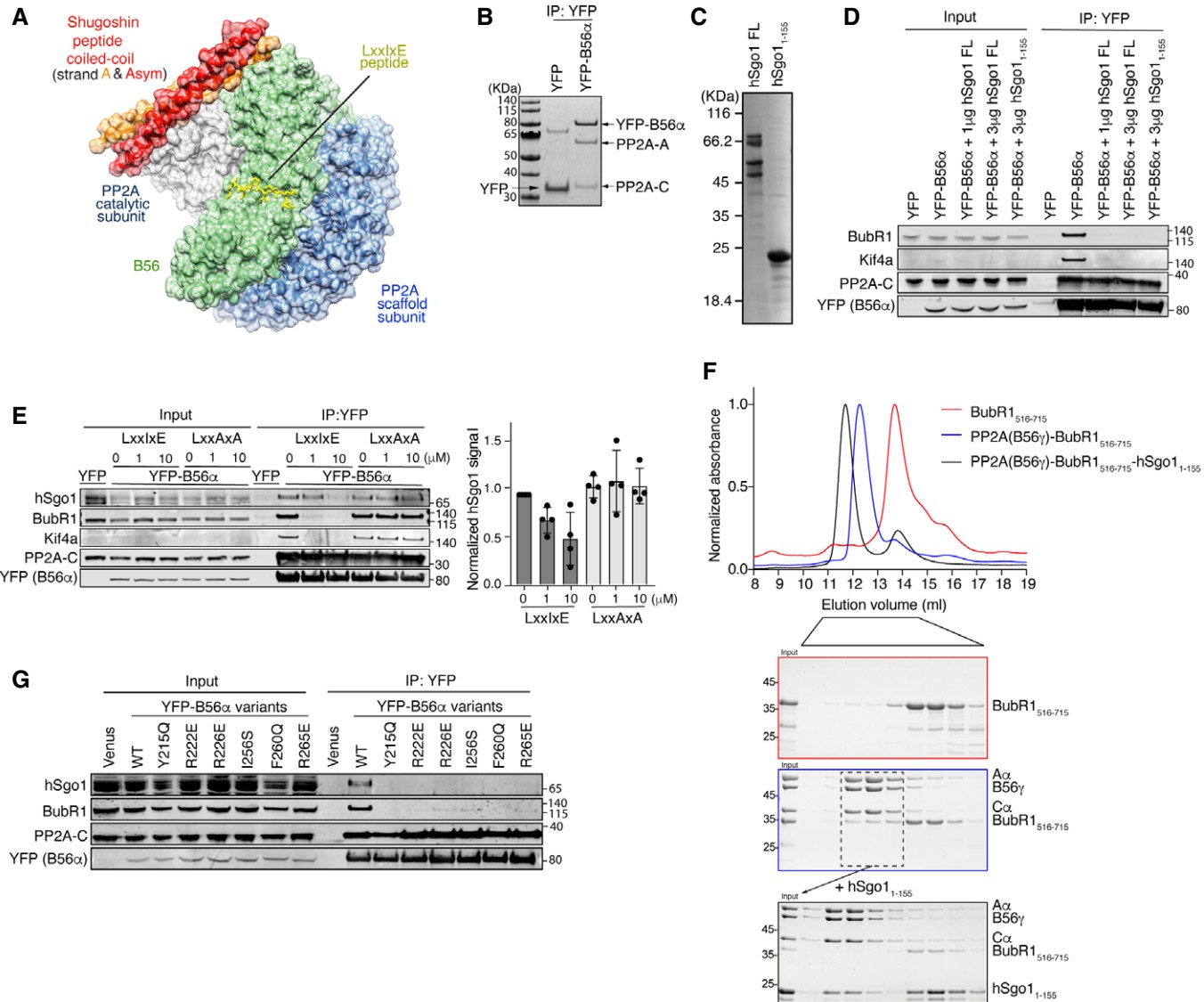

**Figure 1. hSgo1 and LxxIxE motifs compete for binding to PP2A-B56.:**

A   Structure of the PP2A-B56γ–hSgo1 complex (adapted from Xu et al, PDB: 3FGA). The hSgo1 coiled-coil homodimer interacts with both PP2A catalytic and B56 regulatory subunits. The model shows a LxxIxE peptide bound to B56 at its conserved binding pocket.

B   YFP pull down from cells stably expressing YFP (control) or YFP-B56α enriches the entire PP2A-B56α holoenzyme on the beads. PP2A-A, scaffold subunit; PP2A-C, PP2A catalytic subunit.

C   Coomassie-stained SDS–PAGE of the purified hSgo1 full length (FL) and hSgo1$_{1-155}$.

D   Competition assay with the purified hSgo1 proteins shown in (C). Binding of YFP-B56α to indicated proteins was determined. Representative of 3 independent experiments.

E   Peptide competition assay with a WT LxxIxE peptide or a mutated variant that does not bind B56 (LxxAxA). Binding of YFP-B56α to indicated proteins was determined and quantified by LI-COR.

F   A PP2A-B56γ–BubR1$_{516-715}$ complex was reconstituted (blue box) and peak fractions pooled and incubated with hSgo1$_{1-155}$. Following incubation, this complex was analyzed by size exclusion chromatography and fractions analyzed by SDS–PAGE (black box).

G   YFP-B56α pull down from cells stably expressing the indicated LxxIxE binding pocket variants of B56α and subsequent immunoblotting of indicated proteins. Representative of 4 independent experiments.

Source data are available online for this figure.

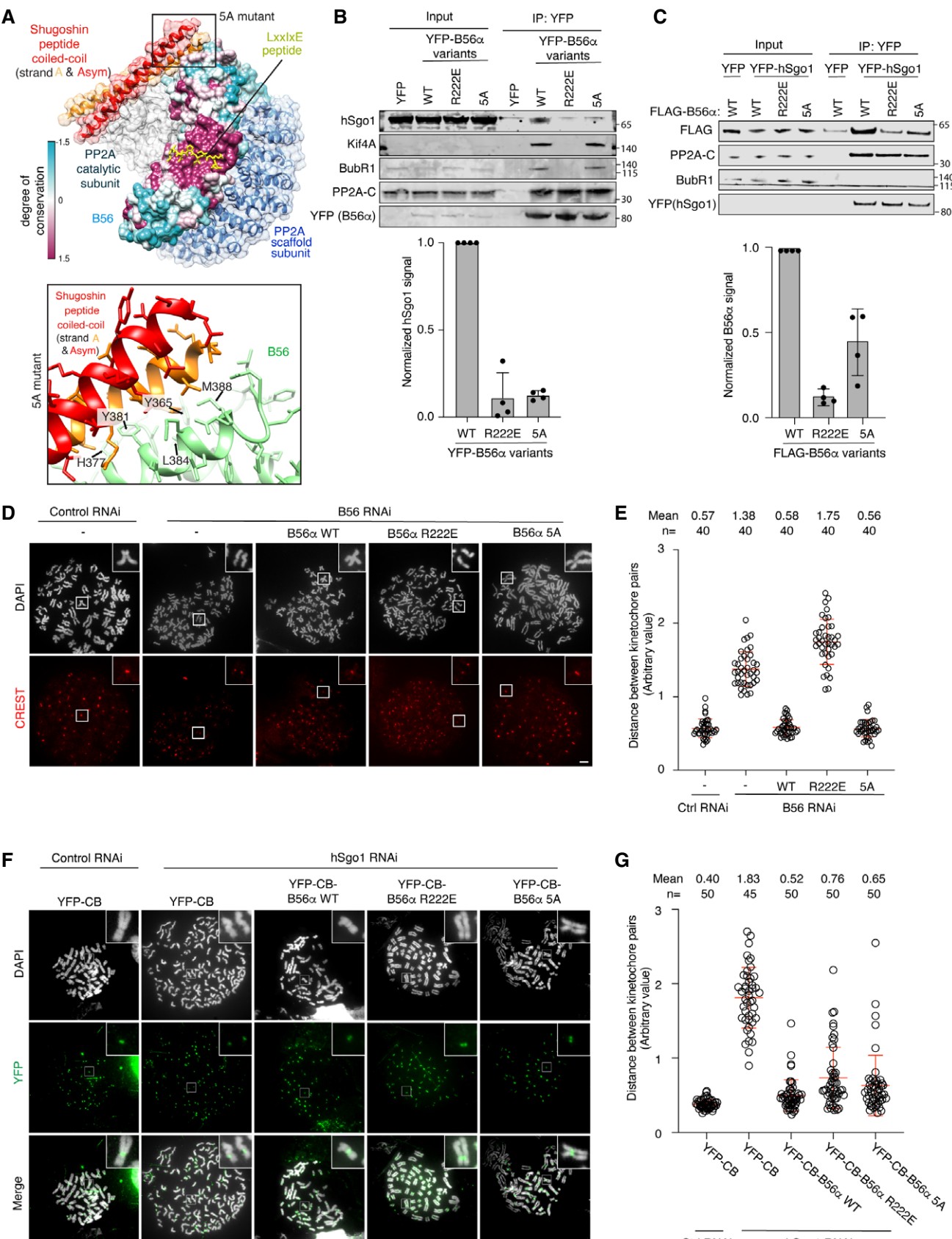

**Figure 2.**

◄

**Figure 2.  hSgo1 binding to the LxxIxE binding pocket of PP2A-B56 is required for cohesion protection.**

A   Structure of the reported PP2Aγ-B56-hSgo1 binding interface (top) and residues mutated in the B56α 5A mutant are shown (bottom).

B   IP of YFP-B56α from cells stably expressing the B56α WT, R222E, and 5A followed by immunoblotting of indicated proteins. Representative blots are shown (top). hSgo1 signals were normalized to YFP and plotted (bottom). Error bars represent SD ($n = 4$).

C   Reciprocal IP of (B). YFP-hSgo1 expression construct was transfected into cells stably expressing FLAG-B56α WT, R222E and 5A, followed by YFP IP and immunoblotting of indicated proteins. Representative blots are shown (top). B56α signals were normalized to YFP and plotted (bottom). Error bars represent SD ($n = 4$).

D   Representative images of chromosome spreads from the indicated conditions. Scale bar, 5 μm.

E   Quantification of (D). The distance between the two peak intensities of CREST was measured for 5 kinetochore pairs and averaged for a single cell and plotted. The data are from 4 independent experiments and the mean and SD are indicated.

F   hSgo1 RNAi and rescue with the indicated B56α variants fused to YFP and the Cenp B centromere-targeting domain (CB). Representative images of chromosome spreads are shown. CB targets all the rescue constructs (green) to the centromere. Scale bar, 5 μm.

G   Quantification of (F). The distance between the two peak intensities of YFP was measured for 5 kinetochore pairs and averaged for a single cell and plotted. The data are from 3 independent experiments and the mean and SD are indicated.

Source data are available online for this figure.

2014). Two proteins, Sororin and the cohesin subunit SA2, have been shown to be dephosphorylated by PP2A-B56-hSgo1 to protect cohesin (Hauf *et al*, 2005; Liu *et al*, 2013; Nishiyama *et al*, 2013). Indeed, expressing variants of Sororin and SA2 that cannot be phosphorylated bypass the need for hSgo1 (Liu *et al*, 2013; Nishiyama *et al*, 2013). Whether PP2A-B56 bound to hSgo1 dephosphorylates other substrates is unclear.

In addition to recruiting PP2A-B56, the shugoshin proteins also recruit the chromosomal passenger complex (CPC) to centromeres through their CC region (Kawashima *et al*, 2007; Vanoosthuyse *et al*, 2007; Tsukahara *et al*, 2010). The shugoshin-dependent localization of CPC to the centromere could also contribute to cohesion protection (Hengeveld *et al*, 2017). Although the interplay between shugoshin recruitment of PP2A-B56 and the CPC to centromeres is not fully established, recent work suggests that the ability of hSgo1/2 to recruit the CPC and PP2A-B56 are distinct activities (Bonner *et al*, 2020).

Crystallographic studies have determined the human PP2A-B56γ in complex with a fragment of hSgo1 comprising residues 51–96, which represents most, but not the entire N-terminal CC domain (Xu *et al*, 2009). This hSgo1 fragment displays less affinity to PP2A-B56γ than longer N-terminal fragments of hSgo1, but sufficient affinity to efficiently bind to PP2A-B56γ under crystallization conditions using high protein concentrations. The structure revealed that the hSgo1 fragment forms a dimer which engages several residues of the last C-terminal HEAT repeat of B56γ and makes contacts to the PP2A catalytic subunit (Fig 1A). Although the crystal asymmetric unit shows a 1:1 interaction between hSgo1 peptide strands and PP2A holoenzymes, the hSgo1 peptide strands are arranged into a parallel CC homodimer, where one fragment is related to the other by a twofold crystallographic symmetry axis (depicted as chain A and Asym in Fig 1A). This arrangement allows them to interact symmetrically with PP2A enzymes on both sides. Thus, one PP2A-B56γ holoenzyme displays interactions with residues from both of the two alpha helices forming one hSgo1 CC region in the crystal, which is again consistent with biochemical experiments showing that dimerization of hSgo1 is required for binding to PP2A-B56γ (Tang *et al*, 1998; Xu *et al*, 2009). In the PP2A-B56γ-hSgo1 structure, the LxxIxE binding pocket of B56γ is fully exposed and indeed the N-terminal region of hSgo1 does not appear to contain any recognizable LxxIxE motif.

These observations raise the possibility that the PP2A-B56-hSgo1 complex can make higher order complexes with LxxIxE containing

proteins, which could be important for mitotic cohesion protection. We explored this possibility, which revealed unanticipated aspects of the PP2A-B56-hSgo1 complex important for understanding cohesion protection during mitosis.

## Results and Discussion

### hSgo1 and LxxIxE motifs compete for binding to PP2A-B56

We first determined whether hSgo1 can bind to PP2A-B56 in complex with LxxIxE containing proteins. We generated stable inducible HeLa cell lines that express YFP-tagged B56α (stable inducible HeLa cell lines used throughout unless indicated) and arrested cells in prometaphase using nocodazole. Mitotic cells were collected by mitotic shake-off, and YFP-B56α was purified using a YFP affinity resin. This enriches the entire PP2A-B56α holoenzyme on the beads (Fig 1B) and co-purifies LxxIxE containing proteins such as BubR1 and Kif4A (Hertz *et al*, 2016). We then incubated the purified YFP-B56α with either recombinantly expressed and purified full-length hSgo1 or an N-terminal fragment of hSgo1 spanning residues 1-155 and washed the complexes (Fig 1C and D). As a control, we treated YFP-B56α purifications with buffer instead of hSgo1. Strikingly, both BubR1 and Kif4a bound to PP2A-B56α in the control samples but were efficiently displaced in the presence of hSgo1 (Fig 1D). We performed a similar experiment in the presence of a high-affinity LxxIxE peptide or the control peptide LxxAxA. The LxxIxE peptide efficiently displaced BubR1 and Kif4A as expected but also reduced hSgo1 binding (Fig 1E). These results suggest that hSgo1 might engage the conserved LxxIxE binding pocket of B56α for binding in cells. To test this with purified components, we reconstituted a PP2A-B56γ-BubR1$_{516-715}$ complex and isolated the complex by size exclusion chromatography. We then incubated this complex with fivefold excess recombinant hSgo1$_{1-155}$ and following incubation characterized the complexes by size exclusion chromatography (Fig 1F). This revealed the formation of a PP2A-B56γ-hSgo1$_{1-155}$ complex devoid of BubR1$_{516-715}$, fully consistent with the cellular data. To further substantiate these results, we used a panel of B56α mutants that have mutations in the LxxIxE binding pocket and analyzed their ability to bind hSgo1. YFP-B56α variants were purified from prometaphase arrested cells, and hSgo1 and BubR1 binding was analyzed. Interestingly, all B56α mutants unable to bind BubR1 failed to co-purify

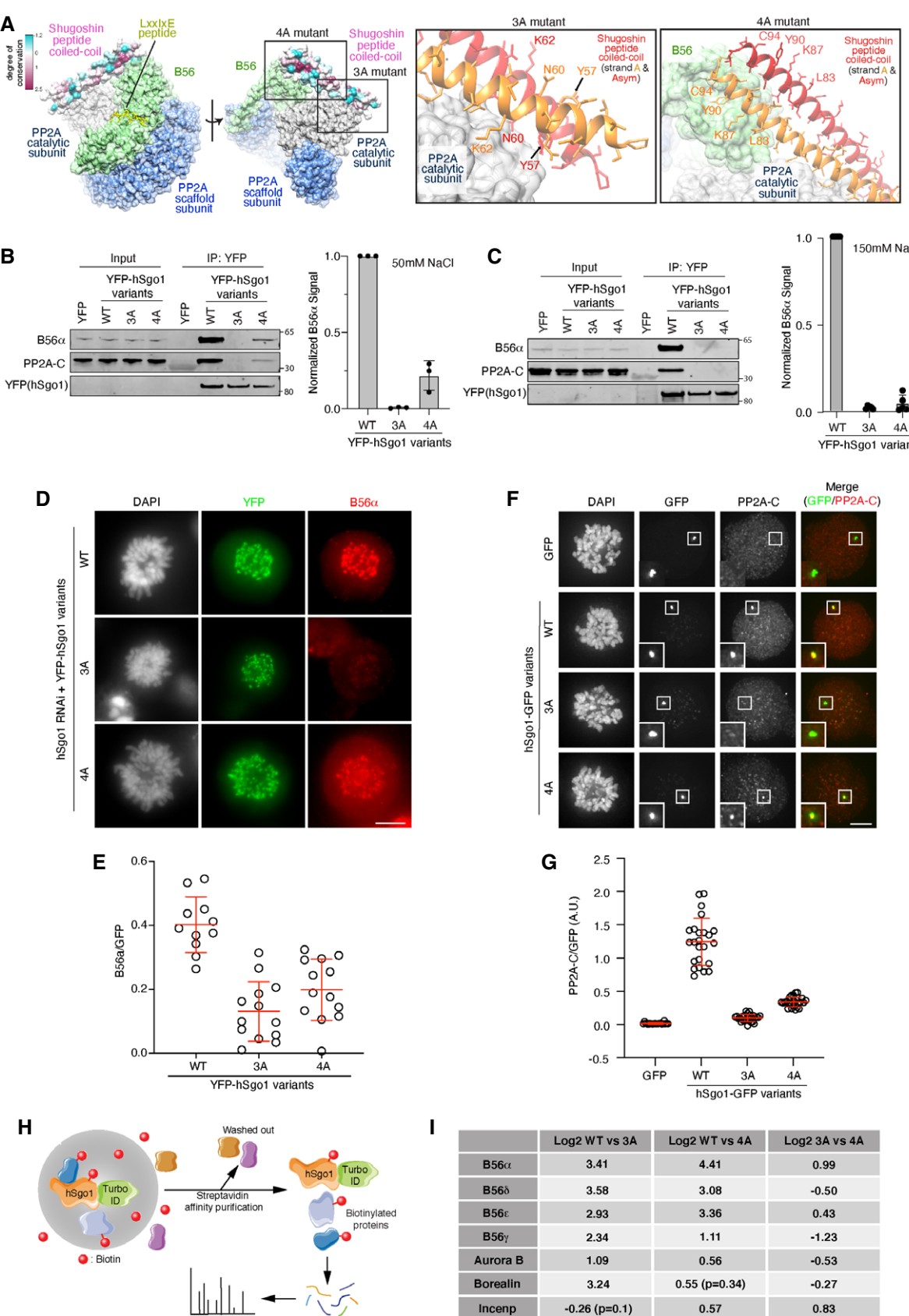

**Figure 3.**

**Figure 3.  hSgo1 mutations in the coiled-coil domain affect PP2A-B56 binding.**

A Structure of the reported PP2A-B56γ-hSgo1 binding interfaces and residues mutated in the hSgo1 3A and 4A mutants are shown. 3A refers to Y57A, N60A, and K62A mutations at the PP2A-C binding interface. 4A refers to L83A, K87A, Y90A, and C94A mutations at the B56 binding interface of hSgo1.

B, C IP of YFP-hSgo1 from cells stably expressing the hSgo1 WT, 3A, and 4A using different salt conditions (50 mM NaCl (B) or 150 mM (C)) followed by immunoblotting of indicated proteins and quantification by LI-COR. Representative blots are shown. B56 signals were normalized to YFP and plotted. Error bars represent SD (*n* = 3 for B and *n* = 4 for C).

D Localization of B56α in cells depleted of hSgo1 and expressing the indicated YFP-hSgo1 variants. Representative immunofluorescent images are shown. Scale bar, 5 μm.

E B56α signal intensity was quantified. B56α signal from each cell was determined from 5 kinetochore pairs and normalized to YFP-hSgo1 signal. Each circle represents an individual cell, and the average and SD are indicated.

F Mitotic U2OS LacO Haspin CM cells expressing hSgo1-LacI-GFP variants or LacI-GFP (control) were stained for PP2A-C. Scale bar, 5 μm.

G PP2A-C signal intensity was quantified, normalized to GFP, and then plotted. Each circle represents an individual cell, and the mean fluorescent intensity is indicated. Representative of at least 3 independent experiments.

H Schematic of the TurboID-hSgo1 approach.

I Table summarizing the Log2 differences between hSgo1 WT, 3A and 4A.

Source data are available online for this figure.

hSgo1 (Fig 1G). The reason why the LxxIxE peptide does not fully displace hSgo1, in contrast to the B56 mutants, could reflect that the PP2A-B56-hSgo1 complex is very stable once formed.

Collectively, these results indicate that LxxIxE motif-containing proteins and hSgo1 compete for a common binding surface on PP2A-B56α. As hSgo1 does not contain any recognizable LxxIxE motif in its CC region, it likely has a binding site that overlaps with the LxxIxE binding pocket of B56 subunits.

**The LxxIxE binding pocket of PP2A-B56 is required for cohesion protection**

The involvement of the B56α LxxIxE binding pocket in hSgo1 binding was surprising, given that hSgo1 binds the less conserved C-terminal HEAT repeat of B56γ in the reported structure of the PP2A-B56γ-hSgo1 complex (Fig 2A). To further analyze this, we investigated the B56α R222E LxxIxE pocket mutant in depth for hSgo1 binding and cohesion protection. We compared this to a B56α mutant (B56α 5A), in which all residues at the reported structural interface with hSgo1 were mutated (B56α 5A:Y365A, H377A, Y381A, L384A, M388A) (Fig 2A). First, we compared the binding of PP2A-B56α to hSgo1 and LxxIxE containing mitotic regulators. Consistent with the reported structure of the PP2A-B56γ-hSgo1 complex, we found that YFP-B56α 5A bound less hSgo1 while maintaining its interactions with BubR1 and Kif4A (Fig 2B). In contrast, B56α R222E (mutation in the LxxIxE binding pocket) lost both binding to hSgo1 and LxxIxE containing proteins. In a reciprocal experiment, cells stably expressing FLAG-tagged B56α variants were transfected with YFP-hSgo1, and then, YFP-hSgo1 was affinity-purified from mitotic cells. Again, we observed impaired binding to both B56α R222E and 5A, with the latter mutant retaining more binding to hSgo1 (Fig 2C). A similar result was obtained using YFP-hSgo2 (Fig EV1A). These experiments strengthen the conclusion that the LxxIxE binding pocket of B56 is an important binding determinant for the human shugoshin proteins.

We next analyzed the ability of the B56α mutants to support cohesion protection. All B56 isoforms were depleted by RNAi and cells were induced to express RNAi-resistant YFP-B56α variants at endogenous levels (Fig EV1B). This in our hands did not affect hSgo1 or hSgo2 localization to centromeres although we note that PP2A has been found to affect hSgo1 localization in another study (Tang *et al*, 2006) (Fig EV1C and D). Cells were synchronized in

prometaphase using nocodazole and chromosome spreads were stained with CREST and DAPI to analyze cohesin integrity. The distance between the two peak intensities of CREST was measured, as premature cohesin removal results in longer distances. Indeed, depleting all B56 subunits increased the distance between centromeres, which was rescued by expressing B56α wild type (WT) (Fig 2D and E). As anticipated from the interaction studies, B56α R222E did not support cohesion protection at all while B56α 5A surprisingly did (Fig 2D and E). To further substantiate this result, we performed live cell imaging of the same conditions. Removing hSgo1 and consequently centromeric cohesin results in prolonged mitotic arrest because of activation of the spindle assembly checkpoint. Similarly, depleting all B56 isoforms resulted in a prolonged arrest which was rescued by YFP-B56α WT and 5A but not the R222E mutant, thus paralleling the chromosome spread results (Fig EV1E–G). Consistent with our binding experiments (Fig 2B and C), only YFP-B56α WT displayed clear localization to chromosomes as observed by live cell imaging (Fig EV1F and Movies EV1–EV3). We analyzed the YFP-B56α 5A phenotype over a range of expression levels and even low levels of expression rescued the B56 RNAi. These results show that mutating the LxxIxE binding pocket of B56α abolishes cohesion protection while the reported interface for binding the hSgo1 CC appears less critical for this.

To establish that B56α R222E can assemble an active PP2A holoenzyme capable of cohesion protection, we artificially recruited the B56α mutants to the centromere by fusing them to the centromere-targeting domain of Cenp B (CB). We then asked if in the absence of hSgo1, these B56α mutants supported cohesion protection (Fig EV2A and B for hSgo1 depletion). We performed chromosome spreads and measured the distance between CREST peak intensities. All variants of CB-B56α rescued the cohesion defect when hSgo1 was depleted, arguing that they form functional PP2A complexes (Fig 2F and G).

Collectively, our analysis of B56α R222E shows that this mutant is defective in cohesion protection most likely due to a defect in hSgo1 binding.

**hSgo1 mutations affecting PP2A-B56 binding**

We were surprised by the fact that the B56α 5A mutant fully supported cohesion protection despite showing a clear reduction in hSgo1 binding. To explore this further, we investigated the

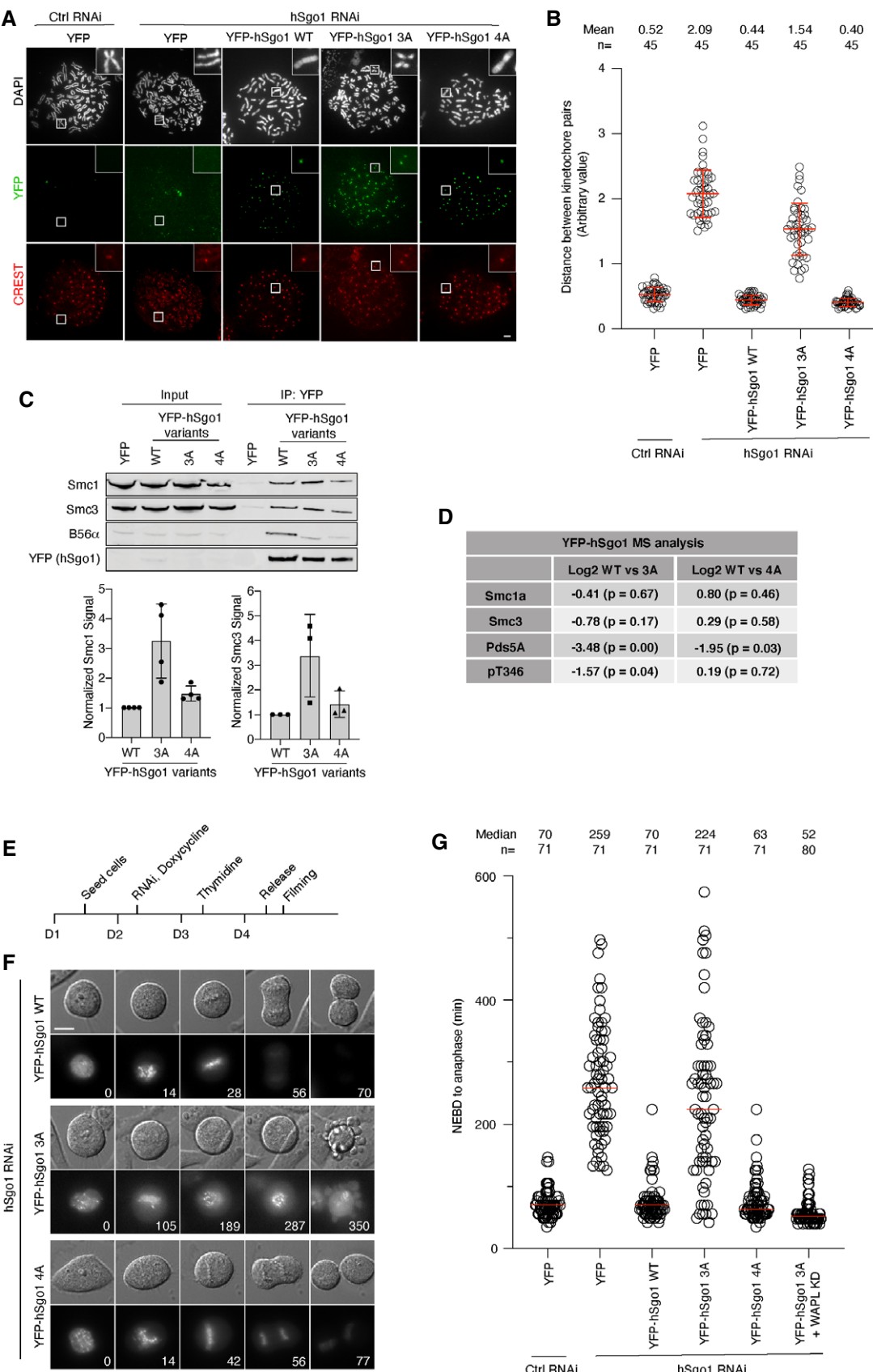

Figure 4.

**Figure 4. Cohesion protection in hSgo1 mutants.**

A   Representative images of chromosome spreads from hSgo1 RNAi treated cells stably expressing the indicated YFP-hSgo1 variants. All the YFP-hSgo1 rescue constructs (green) localize to the centromeres (CREST, red). Scale bar, 5 μm.

B   The distance between the two peak intensities of CREST was measured for 5 kinetochore pairs from the chromosome spreads in (A) and averaged for a single cell. The data are from 3 independent experiments and the mean and SD are indicated.

C   IP of YFP-hSgo1 from cells stably expressing hSgo1 WT, 3A, and 4A followed by immunoblotting of cohesin components, Smc1 and Smc3. Representative blots are shown (top). Signals were quantification by LI-COR, normalized to YFP and plotted (bottom). Error bars represent SD ($n = 4$ for Smc1 and $n = 3$ for Smc3).

D   Table summarizing the Log2 differences between hSgo1 WT, 3A and 4A from quantitative MS analysis.

E   Experimental protocol of the live cell imaging shown in (F).

F   hSgo1 RNAi and rescue with the indicated hSgo1 RNAi-resistant constructs was performed. Representative still images captured during the live cell imaging showing DIC and YFP-hSgo1 WT, 3A, and 4A localization during mitosis. Time (min) from nuclear envelop breakdown (NEBD) is indicated. Scale bar, 15 μm.

G   The time from NEBD to anaphase was measured from 3 independent live cell imaging experiments. Each circle represents an individual cell, and the median is indicated. Note that the WAPL KD condition is incorporated from the experiment which is shown in full in Fig EV4C for clarity.

Source data are available online for this figure.

consequence of mutating the residues in hSgo1 involved in binding the C-terminal region of B56. We generated a hSgo1 mutant (hSgo1 4A) where the four residues (L83, K87, Y90, and C94) contacting B56γ in the reported structure were mutated to alanine residues (Figs 3A and EV2C). As a comparison, we used a previously reported hSgo1 3A mutant (Y57, N60, and K62 to alanine) which contains three mutated residues at the interface with the PP2A catalytic subunit (Xu *et al*, 2009). The reported interface with the PP2A catalytic subunit involves residues from both alpha helices of the hSgo1 CC region (Fig 3A). Stably expressed YFP-hSgo1 variants were purified from mitotic cells at two salt concentrations (50 and 150 mM NaCl), and binding to PP2A-B56α was determined (Fig 3B and C). Both hSgo1 4A and 3A showed a strong reduction in binding to PP2A-B56 components, and only hSgo1 4A maintained some residual binding at the low salt concentration (Fig 3B).

To analyze PP2A-B56-hSgo1 complex formation in cells, we took three separate approaches. Firstly, we analyzed centromeric B56α levels by immunofluorescence in cells depleted of hSgo1 and complemented with RNAi-resistant YFP-hSgo1 variants. By normalizing the B56α signal to YFP, it was clear that both hSgo1 3A and hSgo1 4A localized B56α less efficiently to centromeres (Fig 3D and E). However, it was evident that hSgo1 3A was more compromised in localizing B56α than hSgo1 4A, consistent with the binding assays. Secondly, we employed an assay where LacI-GFP fusions of hSgo1 variants are localized to a LacO array on chromosome 1 in U2OS cells, which allows visualization of PP2A-B56 and CPC recruitment (Figs 3F and G, and EV3). Mitotic cells expressing LacI-GFP fusions of hSgo1 full-length protein and 1–130 were stained for PP2A-C or CPC components (Aurora B and Borealin), and signals were quantified and normalized to GFP. Compared to hSgo1 WT, both hSgo1 4A and hSgo1 3A mutants recruited PP2A-C less efficiently (Figs 3F and G, and EV3D and F). Consistent with the low salt purifications and IF data, hSgo1 4A recruited slightly more PP2A-C than hSgo1 3A. In contrast, we observed more subtle variations in recruitment of CPC components in the full-length hSgo1 constructs while there was no difference using the hSgo1 1–130 constructs (Fig EV3A–C, E, G and H).

In a third approach, we fused the TurboID tag to the N-terminus of hSgo1, and following addition of biotin to mitotic cells for 1 h, we enriched biotinylated proteins under stringent purification conditions (Figs 3H and EV4A). Subsequent analysis of samples by label-free quantitative mass spectrometry revealed labeling of PP2A-B56 components and CPC components and other centromeric proteins

(Fig EV4B, Dataset EV1). Using this approach, we compared PP2A-B56 binding between the different hSgo1 variants. This revealed a significant (*P*-value < 0.05, log2 fold change > 1) reduction in biotinylation of B56 subunits in both hSgo1 3A and 4A compared to hSgo1 WT (Fig 3I, Dataset EV1). Consistent with the LacO array results, we observed a reduction in labeling of CPC components Aurora B and Borealin in hSgo1 3A and 4A compared to hSgo1 WT while INCENP labeling was less affected. In particular, we noted a significantly stronger reduction in Borealin labeling in Sgo1 3A (log2 fold change = 3.24, *P* < 0.05) compared to hSgo1 WT and hSgo1 4A. This subunit has been reported to bind hSgo1 directly (Tsukahara *et al*, 2010). Consistent with these results, it has been reported that hSgo1 3A is less efficient in localizing Ipl1 (Aurora B) in budding yeast (Verzijlbergen *et al*, 2014).

Collectively, we characterize two hSgo1 mutants showing a reduction in PP2A-B56 binding.

### Cohesin binding and protection in hSgo1 mutants

To determine the ability of hSgo1 mutants to support cohesion protection, we depleted endogenous hSgo1 and expressed RNAi-resistant hSgo1 mutants. Cells were arrested in mitosis; chromosome spreads were prepared and distances between kinetochore pairs were measured (Fig 4A and B). hSgo1 depletion resulted in complete loss of cohesion, which was rescued by hSgo1 WT and hSgo1 4A but not by hSgo1 3A. The lack of cohesion protection in hSgo1 3A is consistent with data from budding yeast meiosis (Xu *et al*, 2009). This was not due to a defect in cohesin binding, as both hSgo1 3A and hSgo1 4A enriched more cohesin components in YFP purifications as determined by immunoblotting and quantitative mass spectrometry analysis (Fig 4C and D and Dataset EV2). In particular, the hSgo1 3A mutant bound more Smc1 and Smc3 while hSgo1 4A only showed a slight increase in binding. We observed by mass spectrometry that phosphorylated hSgo1 Thr346 levels were increased in hSgo1 3A, paralleling the increase in cohesin binding (Fig 4D). This argues that PP2A-B56 bound to hSgo1 dephosphorylates Thr346 to negatively regulate cohesin binding. To further analyze the hSgo1 3A and 4A phenotypes, we performed a live cell analysis of cells complemented with the different hSgo1 variants and monitored mitotic progression (Fig 4E–G). hSgo1 depletion induced a strong mitotic arrest which was rescued by hSgo1 WT and hSgo1 4A but not by hSgo1 3A, in agreement with the chromosome spread results. From the live cell analysis, it was obvious that

hSgo1 3A expressing cells did not align chromosomes, likely due to loss of cohesion protection (Fig 4F). Consistent with a specific defect in cohesion protection from the prophase pathway, the depletion of WAPL fully suppressed the hSgo1 3A phenotype (Figs 4G and EV4C). We analyzed all hSgo1 variants at a similar fluorescent intensity and over a range of fluorescent intensities, and this revealed that even low levels of hSgo1 4A expression were sufficient to support hSgo1 function. Further reduction in B56 levels by RNAi did not cause a mitotic phenotype in hSgo1 4A (Fig EV4D–F). This suggests that a substantial reduction in PP2A-B56-hSgo1 complex formation is required to prevent cohesion protection.

An important discovery from our work is that the highly conserved LxxIxE binding pocket of B56 subunits is required for hSgo1 and hSgo2 binding and cohesion protection during mitosis in somatic cells. Indeed, our results argue that the interaction between hSgo1 and the LxxIxE binding pocket of B56 is more critical than the reported interaction with the C-terminal HEAT repeats. Whether these observations extend to meiosis and other model organisms is unclear. This was surprising based on the reported structure of the human PP2A-B56γ-hSgo1 complex and the fact that the B56 binding region of hSgo1 and hSgo2 lacks a recognizable LxxIxE motif. One possibility is that the solved structure, which used only a short fragment of hSgo1, does not fully recapitulate the PP2A-B56-hSgo1 complex and crucial aspects of the structure are yet to be uncovered. We anticipate that full-length hSgo1 binds in a manner that overlaps with the LxxIxE binding pocket of B56 and the C-terminal HEAT repeat as reported in the structure. Consistent with this, *in vivo* cross-linking mass spectrometry identified peptides of hSgo1 cross-linked to residues in close proximity to the LxxIxE binding pocket on B56 (Herzog *et al*, 2012). Given the strong conservation of the B56 LxxIxE binding pocket, our results explain why hSgo1 co-purifies with all isoforms of B56 (Kitajima *et al*, 2006). In addition to this, specific sequence elements present in B56α might further favor hSgo1/2 binding (Vallardi *et al*, 2019). An implication from our results is that hSgo1 and LxxIxE motifs compete for binding to PP2A-B56, which could regulate dephosphorylation at centromeres during mitosis.

Furthermore, our analysis shows that a substantial reduction in PP2A-B56-hSgo1 complex formation can be tolerated without affecting cohesin protection. Only when complex formation is strongly prevented, such as with the hSgo1 3A and B56 R222E mutants, is cohesin protection defective. This might not be surprising given that PP2A-B56 is very efficient in dephosphorylating substrates. Our analysis of the phosphorylation patterns of hSgo1 3A also reveals that one target of PP2A-B56 is Cdk1 phosphorylation sites on hSgo1 itself. Indeed, we find that Thr346 is dephosphorylated by PP2A-B56 and that this regulates hSgo1 association with cohesin during mitosis. Whether dynamic phosphorylation-dephosphorylation of Thr346 is required for cohesin protection during mitosis will be important to investigate in future experiments.

# Materials and Methods

### Antibodies and RNAi oligos

Antibodies used in this study were as follows: Rabbit anti-hSgo1 (gift from Dr. Hongtao Yu, 1:200 IF), rabbit anti-hSgo1 (generated in-house, 1:2,000 WB), mouse anti-B56a (BD Biosciences 610615, 1:2,000 WB and 1:200 IF), rabbit anti-GFP (generated in-house, 1:10,000 WB and 1:500 IF), mouse anti-GFP (Roche #11814460001, 1:2,000 WB and 1:200 IF), mouse anti-BubR1(generated in-house, 1:1,000 WB), rabbit anti-Kif4a (Bethyl Laboratories A301-074A, 1:3,000 WT), mouse anti-PP2A-C (Millipore clone 1D6 05-421, 1:1,000 WB and IF), mouse anti-FLAG M2 (Sigma F3165, 1:10,000 WT), human anti-CREST (Antibodies Inc, 1:500 IF), mouse anti-Aurora B (BD Transductions 611083, 1:1,000 IF), rabbit anti-Borealin (gift from Dr. Sally Wheatley), Smc1 (Bethyl Laboratories A300-055A, 1:2,000 WB), Smc3 (Bethyl Laboratories A300-060A, 1:2,000 WB), and GFP-Booster Atto488 (Chromotek gba488-100, 1:1,000 IF).

RNAi oligos used in this study were as follows: B56α (Dharmacon 5525), B56γ (Dharmacon 5527), B56δ (Dharmacon 5528), B56ε (Dharmacon 5529), hSgo1 (Silencer Select siRNA s45600, Thermo Fischer Scientific), and WAPL (Silencer Select siRNA s22948, Thermo Fischer Scientific).

### Cloning

Standard cloning methods were used throughout the study. pcDNA5/FRT/TO vector was used unless otherwise stated. B56α variants were generated in our previous study (Hertz *et al*, 2016). B56α 5A, hSgo1 3A, and hSgo1 4A mutant constructs were synthesized by GeneArt (Thermo Fischer Scientific). BamHI and NotI were used to subclone B56α and hSgo1 constructs with various tags (YFP, FLAG, or TurboID). Full-length hSgo2 was amplified by PCR and inserted in pcDNA5/FRT/TO-YFP vector. For YFP-CenpB-B56α constructs, CenpB domain was amplified by PCR and inserted into pcDNA5/FRT/TO-YFP vector by standard restriction cloning, followed by subcloning of B56α variants into the vector using BamHI/NotI.

### Cell culture

HeLa FRT/T-REx cells (gift from S. Taylor) were used throughout the study, unless otherwise stated. Stable cell lines were generated using the T-REx doxycycline Flip-In system (Invitrogen). For synchronization, 2.5 mM thymidine and 200 ng/μl nocodazole were used.

### Expression and purification of recombinant hSgo1

BL21 (*DE3*) Gold *E. coli* cells expressing hSgo1 FL and truncations (hSgo1$_{1-155}$) were grown at 37°C/200 rpm to an optical density of 1.5 (OD600) and induced overnight at 18°C with 0.35 mM IPTG. Cells were resuspended in lysis buffer containing 20 mM Tris–HCl pH 8, 500 mM NaCl, 1 mM EDTA and supplemented with complete EDTA-free cocktail tablets (1 tablet/50 ml cells; Roche) and 0.01 mg/ml DNase (Sigma) and 1 mM PMSF. The lysate was sonicated at 60% amplitude for 8 min (2s on, 2s off) and centrifuged at approx. 58,000× *g* for 50 min at 4°C, and the protein was batch purified using chitin beads (NEB). Post lysis and high salt chaperone wash, the chitin beads were washed with 3 CV of 20 mM Tris–HCl, 500 mM NaCl, 50 mM DTT and incubated at RT overnight. The next day, the protein was eluted with the lysis buffer without DTT. The elutions were analyzed for protein quality on an SDS–PAGE, and the elutions containing hSgo1 were pooled and dialyzed in 20 mM Tris–HCl pH 8, 125 mM NaCl, 4 mM DTT overnight at 4°C. The

next day, the dialyzed sample was loaded onto a HiTrap Q HP (GE Healthcare) anionic exchange column. The excess DNA contamination was separated from hSgo1 by providing a 50% salt gradient over 20CV in an ÄKTA start system (GE Healthcare). The samples containing hSgo1 were pooled and concentrated, and the pure protein was finally obtained by a final size exclusion chromatography step with the column equilibrated with 20 mM Tris–HCl, 200 mM salt and 5 mM DTT (Superdex 200 Increase 10/300, GE Healthcare). PP2A(B56γ) and $BubR1_{516-715}$ were produced and purified as previously described (Cho & Xu, 2007; Kruse *et al*, 2013).

### Interaction analysis by size exclusion chromatography

The contribution of the PP2A(B56γ) LxxIxE binding pocket to the binding of $BubR1_{516-715}$ and $hSgo1_{1-155}$ was tested with size exclusion chromatography (SEC) using a Superdex 200 Increase 10/300 column (GE Healthcare). The column was previously equilibrated with 25 mM HEPES pH 7.5, 200 mM NaCl, 5 % glycerol, and 2 mM DTT, and the SEC runs were done with a flow rate of 0.5 ml/min at 4°C. Purified PP2A(B56γ) and $BubR1_{516-715}$ were mixed in a 1:5 molar ratio, respectively, and incubated for 30 min at 4°C prior to SEC analysis. Fractions containing the PP2A(B56γ)-$BubR1_{516-715}$ complex were pooled and concentrated using a 30 kDa Vivaspin® 500 centrifugal unit. The PP2A(B56γ)-$BubR1_{516-715}$ complex was then mixed with a five-times molar excess of $hSgo1_{1-155}$, incubated for 30 min at 4°C and analyzed by SEC. Recombinant $BubR1_{516-715}$ was run separately as a control. SEC fractions were analyzed by SDS–PAGE.

### Immunoprecipitation and competition assays

Inducible, stable cell lines expressing indicated YFP-tagged bait were lysed in a low salt lysis buffer (50 mM Tris–HCl pH 7.5, 50 mM NaCl, 1 mM EDTA, 1 mM DDT, 0.1 % NP-40, protease- and phosphatase inhibitors), unless otherwise stated. In some experiments, the same lysis buffer with 150 mM NaCl was used. Lysates were immunoprecipitated with GFP-trap beads (ChromoTek) according to the manufacturer's recommendation. The beads were washed three times with wash buffer (50 mM Tris–HCl pH 7.5, 1 mg/ml BSA, 20 % glycerol, and 1 mM DTT) and eluted in 2× sample buffer. For the peptide competition assays, a peptide containing LxxIxE motif (LPRSSTLPTIHEEEELSLC) or a control mutant peptide that was unable to bind B56 (LPRSSTLPTA-HAEEELSLC) was used. For the competition assay with hSgo1 proteins, purified full-length hSgo1 or $hSgo1_{1-155}$ described above were used. The peptides/proteins were incubated with cell lysates 30 min prior to the addition of GFP-trap beads.

   IP samples were resolved with 4–12 % Bis-Tris gels (Thermo Fischer Scientific) and transferred to PVDF membranes. LI-COR Odyssey imaging system was used for visualization, and signals were quantified using Image Studio software (LI-COR).

### Chromosome spreads

Indicated cells were seeded in a 6-well plate, RNAi knockdown was performed, and cells were synchronized using thymidine followed by nocodazole treatment. 48 h after RNAi transfection, mitotic cells were collected by shake-off. After hypotonic treatment with KCl, cells were spun onto microscopy slides with a Shandon Cytospin

centrifuge (Thermo Fischer), fixed with 4% paraformaldehyde, and then immunocytochemistry was performed. Representative images were taken with a 100× objective on a DeltaVision fluorescent microscope under the same condition. The distance between the two peak intensities of CREST or YFP-CB was measured for 5 kinetochore pairs using ImageJ and averaged for a single cell. At least 45 cells from minimum of 3 independent experiments were analyzed.

### Live cell imaging

Live cell imaging was performed using a DeltaVision fluorescent microscope. Cells were seeded in a 8-well ibidi dish (ibidi) a day before filming, the media was changed to Leibovitz's L-15 (Life Technologies) immediately before the filming. Indicated channels were recorded at 7- to 8-min intervals, and data were analyzed using SoftWoRx (GE Healthcare). The time from nuclear envelope breakdown (NEBD) to anaphase was measured in single cells.

### Immunofluorescence

Cells were seeded in a 6-well plate and treated as indicated. A day before fixing, cells were transferred in an 8-well ibidi dish. Cells were fixed in 4% PFA for 10 min at room temperature, and standard immunocytochemical methods were used. Fluorescent microscopy was performed using a DeltaVision fluorescent microscope. To ensure quantitative image quality, the imaging parameters were kept constant for a given experiment.

### LacO-LacI assay

hSgo1-LacI-GFP constructs were cloned in a pAceBac1-CMV background (Hadders *et al*, 2020). Bacmids were generated using the Bac-to-Bac system in conjunction with EMBacY cells (Berger *et al*, 2004; Bieniossek *et al*, 2012). Baculovirus was then produced by transfection of bacmids into Sf9 cells using standard procedures. P2 viruses were harvested after 5 days, filtered (0.2 µm), and stored at 4°C till use. The lacO-LacI assays were performed as previously described in (Hadders *et al*, 2020). Briefly, U-2 OS LacO Haspin CM (CRISPR Mutant) cells were seeded on glass coverslips followed directly by addition of recombinant baculovirus encoding the hSgo1-LacI-GFP variants or LacI-GFP as a control. After 3–4 h, S-trityl-L-cysteine (STLC; 20 µM) was added overnight to block cells in mitosis. The next morning cells were fixed in 4% PFA (v/v) in PHEM buffer (60 mM HEPES KOH, 20 mM PIPES KOH, pH 6.8, 5 mM EGTA, and 1 mM $MgCl_2$) for 10–15 min followed by permeabilization in ice-cold methanol for a minimum of 1 h.

   For immunofluorescence, cells were washed with PBS with 0.01% Tween-20 (PBST), followed by blocking with 3% BSA in PBST for ± 30 min. Cells were then incubated with primary antibodies diluted in 3% BSA in PBST for 2 h followed by washing three times, again with PBST. Cells were then incubated with secondary antibodies, GFP-Booster, and DAPI (500 ng/µl) in PBST + 3% BSA for 1 h. Coverslips were washed again, twice with PBST, followed by a final wash with PBS, before mounting onto glass slides using Prolong Diamond Antifade Mountant (Thermo Fisher Scientific).

   Fluorescence images were acquired on a DeltaVision imaging system (GE Healthcare), upgraded with a seven-color InsightSSI Module & TruLight Illumination System Module using a UPlanSApo

60×/1.40 objective and a CoolSnap HQ2 camera (Photometrics). 3D z-stacks were collected and deconvolved using Softworx v6. Presented images are deconvolved maximum intensity projections. Quantifications were performed using an in-house-developed macro in ImageJ that sets a threshold (Otsu) based on the GFP channel followed by measurement of all channels within this region of interest.

### TurboID proximity labeling and label-free LC-MS/MS analysis

TurboID proximity labeling assay was performed as described previously (Branon *et al*, 2018). Doxycycline-inducible TurboID-hSgo1 WT, 3A and 4A stable cell lines were generated in HeLa cells, and 50 μM biotin was added to the media 1 h prior to the harvest. Cells were collected by mitotic shake-off, lysed with RIPA buffer, and immunoprecipitation was performed using high-capacity Streptavidin agarose beads (Thermo Scientific). The beads were washed once with RIPA buffer, twice with 2% SDS, then again once with RIPA buffer and eluted in 2× sample buffer. For YFP-Sgo1 WT, 3A, 4A analysis, YFP purifications were performed as described above. The elutes were then run on SDS gels and sliced for MS analysis. Pull-downs were analyzed on a Q-Exactive Plus quadrupole, Fusion Orbitrap, or Fusion Orbitrap Lumos mass spectrometer (Thermo Scientific) equipped with Easy-nLC 1000 or Easy-nLC 1200 (Thermo Scientific) and nanospray source (Thermo Scientific). Peptides were resuspended in 5% methanol / 1% formic acid and analyzed as previously described (Kruse *et al*, 2020). Raw data were searched using COMET (release version 2014.01) in high-resolution mode (Eng *et al*, 2013) against a target-decoy (reversed) (Elias & Gygi, 2007) version of the human proteome sequence database (UniProt; downloaded 2/2020, 40704 entries of forward and reverse protein sequences) with a precursor mass tolerance of ± 1 Da and a fragment ion mass tolerance of 0.02 Da, and requiring fully tryptic peptides (K, R; not preceding P) with up to three mis-cleavages.

Static modifications included carbamidomethylcysteine and variable modifications included oxidized methionine, and phosphorylatedserines, threonines, and tyrosines. Searches were filtered using orthogonal measures including mass measurement accuracy (± 3 ppm), Xcorr for charges from +2 through +4, and dCn targeting a < 1% FDR at the peptide level. Quantification of LC-MS/MS spectra was performed using BasicQuan or MassChroQ (Valot *et al*, 2011) and the iBAQ method (Schwanhausser *et al*, 2011). Missing values were imputed from a normal distribution in Perseus to enable statistical analysis and visualization by volcano plot (Tyanova *et al*, 2016). For phosphorylation site analysis, retention time alignment, and smart quantification was performed. For further analysis, proteins had to be identified in the hSgo1 +biotin or hSgo1 WT samples with more than 1 total peptide and quantified in 2 or more replicates. For further analysis, phosphorylation sites had to identify in at least 5 out 9 samples. hSgo1 amounts were normalized to be even across WT, 3A, and 4A samples. Statistical analysis was carried out in Perseus by two-tailed Student's t-test.

## Data availability

The mass spectrometry data have been deposited to ProteomeXchange PXD024532 and MassIVE MSV000087003. http://proteomecentral.proteomexchange.org/cgi/GetDataset?ID=PXD024532

Expanded View for this article is available online.

## Acknowledgements

Work at the Novo Nordisk Foundation Center for Protein Research is supported by grant NNF14CC0001 and JNI is supported by grants from the Danish Cancer Society (R167- A10951-17-S2), Independent Research Fund Denmark (8021-00101B), and Novo Nordisk Foundation (NNF18OC0053124). This work was also supported by the cryo-EM (NNF0024386) and cryoNET (NNF17SA0030214) grants to G.M. G.M. is a member of the Integrative Structural Biology Cluster (ISBUC) at the University of Copenhagen. M.W. was supported by the Swiss National Fund (P2EZP3_178624) and the Danish Lundbeckfonden (2017-3212). ANK was supported by grants from NIH/NIGMS (R35GM119455). The Orbitrap Fusion Tribrid mass spectrometer was acquired with support from NIH (S10-OD016212). MH and SMAL are part of the Oncode Institute, which is partly financed by the Dutch Cancer Society and work for this project was funded by a grant from the Dutch Cancer Society (KWF research project 10366). AAJ acknowledges the Wellcome Trust for their support through the Research Career Development (095822) and Senior Research Fellowship (202811) awards. TG acknowledges the Darwin Trust of Edinburgh for the studentship. We thank Hongtao Yu for providing Sgo1 antibody and Sally Wheatley for providing the Borealin antibody. We thank Adele Marston for comments on the manuscript and fruitful discussions. We thank the protein production facility at NNF CPR for help with producing recombinant BubR1.

## Author contributions

YU did all experiments except as indicated below. Experiment in Fig 1F which was performed by PP-S, TG, and AAJ including production of reconstituted complexes and hSgo1. All lacI array experiments were performed and analyzed by MAH and SMAL. Structural analysis of the PP2A-B56-hSgo1 complex was done by MBW and GM. All MS analysis was performed by IN, LEC, and AK. EPTH and TK generated initial reagents and data for the project. JN managed the project acquired funding and wrote the initial draft of the manuscript. All authors contributed to discussion of data and editing the final manuscript.

## Conflict of interest

GM is a co-founder and board member of Twelve Bio. The rest of the authors declare that they have no conflict of interest.

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
