## [Review Process File · EMBO Reports]

A highly conserved pocket on PP2A-B56 is required for hSgo1 binding and cohesion protection during mitosis

Yumi Ueki, Michael Hadders, Melanie Weisser, Isha Nasa, Paula Sotelo-Parrilla, Lauren Cressey, Tanmay Gupta, Emil Hertz, Thomas Kruse, Guillermo Montoya, A. Arockia Jeyaprakash, Arminja Kettenbach, Susanne Lens, and Jakob Nilsson

DOI: [10.15252/embr.202052295](https://doi.org/10.15252/embr.202052295)

Corresponding author(s): Jakob Nilsson (jakob.nilsson@cpr.ku.dk)

Review Timeline:

Submission Date:	17th Dec 20
Editorial Decision:	28th Jan 21
Revision Received:	10th Mar 21
Editorial Decision:	6th Apr 21
Revision Received:	7th Apr 21
Accepted:	13th Apr 21

Editor: Deniz Senyilmaz Tiebe

Transaction Report:

Dear Jakob,

Thank you for submitting your manuscript to EMBO Reports. Three referees agreed to review your manuscript. So far, we have received two referee reports that are copied below. Given that both referees are in fair agreement that you should be given a chance to revise the manuscript, I would like to ask you to begin revising your study along the lines suggested by the referees.

I apologize for this unusual delay in getting back to you. It took longer than anticipated to receive the referee reports due to the recent holiday season.

Please note that this is a preliminary decision made in the interest of time, and that it is subject to change should the third referee offer very strong and convincing reasons for this. As soon as we receive the final report on your manuscript, we will forward it to you as well.

Referees express interest in the proposed mechanism of Sgo mediated cohesion protection. However, they also raise important concerns that need to be addressed to consider publication here.

I find the reports informed and constructive, and believe that addressing the concerns raised will significantly strengthen the manuscript. As the reports are below, and I think all points need to be addressed, I will not detail them here.

Given these constructive comments, we would like to invite you to revise your manuscript with the understanding that the referee concerns (as in their reports) must be fully addressed and their suggestions taken on board. Please address all referee concerns in a complete point-by-point response. Acceptance of the manuscript will depend on a positive outcome of a second round of review. It is EMBO reports policy to allow a single round of revision only and acceptance or rejection of the manuscript will therefore depend on the completeness of your responses included in the next, final version of the manuscript.

*** Temporary update to EMBO Press scooping protection policy:

We are aware that many laboratories cannot function at full efficiency during the current COVID-19/SARS-CoV-2 pandemic and have therefore extended our 'scooping protection policy' to cover the period required for a full revision to address the experimental issues highlighted in the editorial decision letter. Please contact the scientific editor handling your manuscript to discuss a revision plan should you need additional time, and also if you see a paper with related content published elsewhere.***

IMPORTANT NOTE: we perform an initial quality control of all revised manuscripts before re-review. Your manuscript will FAIL this control and the handling will be DELAYED if the following APPLIES:
1. A data availability section providing access to data deposited in public databases is missing

(where applicable).

2. Your manuscript contains statistics and error bars based on $n=2$. Please use scatter plots in these cases.

Supplementary/additional data: The Expanded View format, which will be displayed in the main HTML of the paper in a collapsible format, has replaced the Supplementary information. You can submit up to 5 images as Expanded View. Please follow the nomenclature Figure EV1, Figure EV2 etc. The figure legend for these should be included in the main manuscript document file in a section called Expanded View Figure Legends after the main Figure Legends section. Additional Supplementary material should be supplied as a single pdf labeled Appendix. The Appendix includes a table of content on the first page with page numbers, all figures and their legends. Please follow the nomenclature Appendix Figure Sx throughout the text and also label the figures according to this nomenclature. For more details please refer to our guide to authors.

Please note that for all articles published beginning 1 July 2020, the EMBO Reports reference style will change to the Harvard style for all article types. Details and examples are provided at <https://www.embopress.org/page/journal/14693178/authorguide#referencesformat>

2) individual production quality figure files as .eps, .tif, .jpg (one file per figure).

3) a .docx formatted letter INCLUDING the reviewers' reports and your detailed point-by-point responses to their comments. As part of the EMBO Press transparent editorial process, the point-by-point response is part of the Review Process File (RPF), which will be published alongside your paper. For more details on our Transparent Editorial Process, please visit our website: <https://www.embopress.org/page/journal/14693178/authorguide#transparentprocess>
You are able to opt out of this by letting the editorial office know (emboreports@embo.org). If you do opt out, the Review Process File link will point to the following statement: "No Review Process File is available with this article, as the authors have chosen not to make the review process public in this case."

4) a complete author checklist, which you can download from our author guidelines (<http://embor.embopress.org/authorguide>). Please insert information in the checklist that is also reflected in the manuscript. The completed author checklist will also be part of the RPF.

5) Please note that all corresponding authors are required to supply an ORCID ID for their name upon submission of a revised manuscript (<https://orcid.org/>). Please find instructions on how to link your ORCID ID to your account in our manuscript tracking system in our Author guidelines (<http://embor.embopress.org/authorguide>).

6) We replaced Supplementary Information with Expanded View (EV) Figures and Tables that are collapsible/expandable online. A maximum of 5 EV Figures can be typeset. EV Figures should be cited as 'Figure EV1, Figure EV2' etc... in the text and their respective legends should be included in the main text after the legends of regular figures.

- For the figures that you do NOT wish to display as Expanded View figures, they should be bundled together with their legends in a single PDF file called *Appendix*, which should start with a short Table of Content. Appendix figures should be referred to in the main text as: "Appendix Figure S1, Appendix Figure S2" etc. See detailed instructions regarding expanded view here: <<http://embor.embopress.org/authorguide#expandedview>>.

7) We would also encourage you to include the source data for figure panels that show essential data.

Numerical data should be provided as individual .xls or .csv files (including a tab describing the data). For blots or microscopy, uncropped images should be submitted (using a zip archive if multiple images need to be supplied for one panel). Additional information on source data and instruction on how to label the files are available <<http://embor.embopress.org/authorguide#sourcedata>>.

8) Our journal encourages inclusion of *data citations in the reference list* to directly cite datasets that were re-used and obtained from public databases. Data citations in the article text are distinct from normal bibliographical citations and should directly link to the database records from which the data can be accessed. In the main text, data citations are formatted as follows: "Data ref: Smith et al, 2001" or "Data ref: NCBI Sequence Read Archive PRJNA342805, 2017". In the Reference list, data citations must be labeled with "[DATASET]". A data reference must provide the database name, accession number/identifiers and a resolvable link to the landing page from which the data can be accessed at the end of the reference. Further instructions are available at <<http://embor.embopress.org/authorguide#datacitation>>.

9) Please make sure to include a Data Availability Section before submitting your revision - if it is not applicable, make a statement that no data were deposited in a public database. Primary datasets (and computer code, where appropriate) produced in this study need to be deposited in an appropriate public database (see <<http://embor.embopress.org/authorguide#dataavailability>>).

The accession numbers and database should be listed in a formal "Data Availability " section (placed after Materials & Method) that follows the model below. Please note that the Data Availability Section is restricted to new primary data that are part of this study.

Data availability

10) Regarding data quantification, please ensure to specify the name of the statistical test used to generate error bars and P values, the number (n) of independent experiments underlying each data point (not replicate measures of one sample), and the test used to calculate p-values in each figure legend. Discussion of statistical methodology can be reported in the materials and methods section, but figure legends should contain a basic description of n, P and the test applied.

Please note that error bars and statistical comparisons may only be applied to data obtained from at least three independent biological replicates.

I look forward to seeing a revised version of your manuscript when it is ready. Please let me know if you have questions or comments regarding the revision.

Kind regards,

Deniz

Deniz Senyilmaz Tiebe, PhD
Editor
EMBO Reports

Referee #1:

This manuscript deals with the role of Shugoshin 1 (Sgo1) in centromeric cohesin protection, which is (partially) mediated by dephosphorylation of Sororin and the cohesin subunit SA2 by Sgo1-recruited PP2A-B56. Previous crystallisation studies with a Sgo1 fragment have identified a coiled-coil-region that binds to the last HEAT repeat of the B56 subunit of PP2A-B56. The present manuscript shows that this structure is incomplete and demonstrates by mutagenesis and competition studies that Sgo1 also interacts with a previously characterized substrate-docking LxxxE-motif of the B56 subunit, a motif that is important for maintaining mitotic cohesion. The authors furthermore find that B56-binding mutants of Sgo1 still support mitotic cohesion, suggesting that cohesion protection by Sgo1 is at least partially independent of PP2A-B56 recruitment.

Overall, this is an interesting manuscript on a complicated topic. The experiments are carefully designed and executed, the data are clean and correctly interpreted, and the manuscript is well written.

Specific comments

Since Sgo1 does not have a recognizable LxxxE-motif it seems unlikely that it binds to B56 in an identical manner as other proteins with a canonical LxxxE motif. It could be argued that Sgo1 has an overlapping, but not identical binding site.

Is cohesion protection by B56-binding mutants of Sgo1 explained by competition with WAPL or by recruitment of the CPC?

Referee #2:

Ueki et al investigate the mechanism of Sgo mediated cohesion protection. Sgo1 protects cohesion by binding of cohesion via a conserved binding motif facilitating the dephosphorylation of Sororin and SA2. Furthermore, Sgo1 competes with WAPL for cohesin binding. Previous structural studies using a Sgo1 fragment comprising most of the N-terminal coiled coil domain revealed that it binds as a dimer to the last C-terminal heat repeat of B56gamma and makes contacts to the catalytic subunit. Notably, the LxxlxE binding pocket is fully exposed under these conditions suggesting that it does not contribute to the binding of Sgo1 to B56gamma. Unexpectedly, pull down assays revealed that Sgo1 seems to compete with bona fide LxxlxE containing proteins such as Kif4A and BubR1 for B56 binding. This finding was further supported by B56alpha mutants. Both, the B56 5A as well as the R222E mutant showed reduced Sgo1 binding. Surprisingly, RNAi rescue experiments revealed that the 5A, but not the R222E mutant, supports cohesion protection despite its reduced capacity to bind Sgo1. To support this finding in a complementary manner, Ueki et al introduced mutations in Sgo1. Both, Sgo1 4A and 3A mutants showed reduced/no binding to B56 and PP2A-C. Lac/LacO-based experiments confirmed reduced recruitment of PP2A-C in the case of the 3A and 4A mutants. In cells, Sgo1 4A mutant, but not the 3A mutant, rescued cohesion protection. Even in the presence of reduced B56 levels, the 4A mutant was capable of rescuing mitotic timing.

This ms describes an interesting finding which actually raises more new questions than answering old ones. This reviewer thinks that the manuscript should be published once the authors address the following points.

The Sgo1 4A mutant - despite its significant binding capacity to B56 - rescues cohesion protection. Based on this observations, the authors speculate in the discussion that either low levels of Sgo1 binding to PP2A-B56 is sufficient or binding of PP2A-B56 to Sgo1 is not required for cohesion protection. Since partial depletion of B56 did not affect mitotic timing of the 4A mutant compared to WT Sgo1, the authors argue that binding of PP2-B56 to Sgo1 might not be required for cohesion protection. To make this statement the authors have to repeat the 4A pulldown in cells partially depleted of B56 in order to confirm that under these conditions less B56 binds to the 4A mutant.

Fig. 3 D and F: According to the immunofluorescence in 3D, the 4A mutant recruits much more PP2A-C than the 3A mutant. According to 3F, the difference is minor. Could the authors please address this issue. Also, the authors should quantify B56 localization in this setup.

Fig. EV1F: This reviewer cannot see chromosome localization of WT B56alpha.

Previously, it has been shown that Cdk1 phosphorylation of Sgo1 is essential for cohesin binding and its cohesion protective function. The Sgo1 T346 non-phosphorylatable mutant binds PP2A and localizes to centromeres, but fails to bind to cohesin. Since the authors address the role of the Sgo1-PP2A-B56 axis in cohesion protection, this reviewer would have liked to see more experiments in this direction, e.g. how do the different Sgo1 and B56 mutants analyzed in this study behave with respect to cohesin binding. This is an important point.

Dear Jakob,

We have now received the report of referee #3 and I am copying it below as mentioned in my previous letter. Please respond to the comments of this referee as well during the revision.

I am looking forward to reading your revised manuscript when it is ready. Please contact us if you have any questions or comments about the revision.

Kind regards,

Deniz

--

Deniz Senyilmaz Tiebe, PhD

Editor

EMBO Reports

Referee #3:

The authors performed a structure-function analysis of the interaction of hSgo1 (Sgol1) with PP2A-B56. The analysis was guided by a previously published crystal structure of PP2A-B56 bound to a fragment of hSgo1 and recent progress in our understanding of substrate recognition by PP2A-B56 pioneered by the Nilsson lab.

The authors provide *in vitro* and *in vivo* evidence that Sgol1 competes with LxxIxE-containing proteins for binding to B56. Accordingly, a mutation (R222E) in B56's LxxIxE-binding pocket reduces binding of Sgol1 and other LxxIxE-containing proteins. The R222E mutation increases the interkinetochore distance in nocodazole-arrested cells, an assay for protection of centromeric cohesion in mitosis. These data suggest that B56's LxxIxE-binding pocket is important for binding to Sgol1, which is, in turn, required for protection of centromeric cohesion from the prophase pathway in human cells in culture. Whether Sgol1 actually has a functional LxxIxE motif has not been investigated.

The crystal structure published by Xu et al. revealed an interaction between Sgol1's CC domain and a surface involving residues of B56 and PP2A-C. The authors investigated B56's interaction with Sgol1 by creating mutation in B56 (B56-5A) and Sgol1. The latter mutant is called Sgol1-4A, while the mutant of Xu et al. in Sgol1 residues contacting PP2A-C is known as Sgol1-3A. B56-5A and Sgol1-4A reduce the B56-Sgol1 interaction *in vitro* and *in vivo* but less so than Sgol1-3A. While the Sgol1-3A mutation abolished protection of centromeric cohesion, the mutations B56-5A and Sgol1-4A have not detectable effect. These results let the authors favor the hypothesis that Sgol1 and PP2A-B56 have independent functions in the protection of centromeric cohesion from the prophase pathway.

Major points

Whereas I am reluctant to ask for additional experiments, I feel that the text of the manuscript has to be revised before considering publication.

1. The finding that B56's LxxIxE-binding pocket is required for Sgol1 binding and protection of centromeric cohesion from the prophase pathway is important for

researchers working on the function and regulation of shugoshins and centromeric cohesion in mitosis and meiosis. Given that there is no analysis of a potential LxxIxE motif in Sgol1, it would be nice if the authors could at least point out candidate motifs in the Sgol1 sequence. The Nilsson group is arguably in the best position to evaluate the potential of more derived motifs.

2. Admittedly, the literature on shugoshin is huge and the nomenclature of shugoshins in different organisms is somewhat confusing and inconsistent. It would be beneficial, therefore, if the authors could point out early on that there are two shugoshins in mammals: Sgo1/Sgol1 protects centromeric cohesion from the (non-proteolytic) prophase pathway in mitotic cells, while Sgo2/Sgol2 protects centromeric Rec8 from cleavage by separase in meiosis. In yeast, it is Sgo1 that protects Rec8 in meiosis. So, the statement at the end of the first paragraph is correct for yeast but incorrect for mammals. Since this work is on mammalian Sgol1, this creates confusion for the "untrained eye". In fact, the authors might consider calling "their" Sgo1 either hSgo1 or Sgol1 to clearly distinguish it from the yeast proteins, which are relevant to this work as some implication of the Xu et al. crystal structure have been tested in yeast.

3. In the same vein, it would be desirable to more precisely state in the title and discussion what has been investigated here: a structure-function analysis of the interaction of Sgol1 with PP2A-B56 in human cells in culture and the implications thereof for protection of centromeric cohesion from the prophase pathway. Surprisingly, the prophase pathway has not been mentioned at all in the manuscript. To what extent these conclusions are applicable to other shugoshins or to protection of centromeric Rec8 in meiosis can or should be discussed. Ultimately, however, this remains to be investigated.

4. The citations at the end of the first paragraph should include Kitajima et al., 2006 (the Nature paper showing that centromeric Rec8 is protected by Sgo1-PP2A-B56 in fission yeast).

5. The authors find that the B56-5A and Sgol1-4A mutations weaken the Sgol1-PP2A-B56 interaction but do not seem to affect protection of centromeric cohesion. Based on this finding they promote the hypothesis that Sgol1 and PP2A-B56 have independent functions in protection. In my opinion, at least, the data do not justify the room given to this hypothesis. The data merely suggest that the protection mechanism tolerates a weakening of the interaction, which is consistent with the findings that very small amounts of Sgol1-PP2A in mammals or Sgo1-PP2A in yeast are sufficient for protection of centromeric cohesion. It is not uncommon that a reduction in the affinity of a protein-protein interaction has little effect on the process in the living cells where protein concentrations can be very high, especially for proteins accumulating at a particular structure (like the kinetochore or centromere). The authors used a similar argument for binding of the Sgol1 fragment to PP2A-B56 under crystallization conditions.

6. The last part of the discussion is a bit confusing and reveals a rather lax interpretation of the literature. The finding that the Sgol1-3A mutation affects several proteins is not surprising given the multitude of functions of shugoshin proteins at kinetochores/centromeres. Also, the comments with regards to expressing of soronin or Rec8 mutants are inaccurate. For instance, the Rec8

mutants, the corresponding kinase mutants, Sgo1 mutants, etc have been expressed at endogenous levels (replacing the wild-type genes) and to retain other function, such as providing cohesion or protein interactions. The authors risk, unnecessarily in my opinion, confusing the general reader or antagonizing researchers in the shugoshin field.

Point to point response**Referee #1:**

This manuscript deals with the role of Shugoshin 1 (Sgo1) in centromeric cohesin protection, which is (partially) mediated by dephosphorylation of Sororin and the cohesin subunit SA2 by Sgo1-recruited PP2A-B56. Previous crystallisation studies with a Sgo1 fragment have identified a coiled-coil-region that binds to the last HEAT repeat of the B56 subunit of PP2A-B56. The present manuscript shows that this structure is incomplete and demonstrates by mutagenesis and competition studies that Sgo1 also interacts with a previously characterized substrate-docking LxxIxE-motif of the B56 subunit, a motif that is important for maintaining mitotic cohesion. The authors furthermore find that B56-binding mutants of Sgo1 still support mitotic cohesion, suggesting that cohesion protection by Sgo1 is at least partially independent of PP2A-B56 recruitment.

Overall, this is an interesting manuscript on a complicated topic. The experiments are carefully designed and executed, the data are clean and correctly interpreted, and the manuscript is well written.

We thank the reviewer for the positive comments.**Specific comments**

Since Sgo1 does not have a recognizable LxxIxE-motif it seems unlikely that it binds to B56 in an identical manner as other proteins with a canonical LxxIxE motif. It could be argued that Sgo1 has an overlapping, but not identical binding site.

Our response:

We agree with the reviewer on this. Sgo1 must have an overlapping binding mechanism and not a similar binding mechanism. We have indicated this in the revised manuscript at the end of the first result section as well as the discussion.

Is cohesion protection by B56-binding mutants of Sgo1 explained by competition with WAPL or by recruitment of the CPC?

Our response:

We thank the reviewer for suggesting this experiment. We can fully rescue the Sgo1 3A phenotype by depleting WAPL suggesting that the main defect of Sgo1 3A is to counteract WAPL activity. These data are incorporated into the revised manuscript as Figure 4G. We note that Sgo1 RNAi cannot be suppressed by WAPL RNAi in live cell imaging experiments suggesting that Sgo1 must have functions unrelated to PP2A-B56 binding that is required for alignment of chromosomes.

This could possibly be related to CPC localization by Sgo1.

Referee #2:

Ueki et al investigate the mechanism of Sgo mediated cohesion protection. Sgo1 protects cohesion by binding of cohesion via a conserved binding motif facilitating the dephosphorylation of Sororin and SA2. Furthermore, Sgo1 competes with WAPL for cohesin binding. Previous structural studies using a Sgo1 fragment comprising most of the N-terminal coiled coil domain revealed that it binds as a dimer to the last C-terminal heat repeat of B56gamma and makes contacts to the catalytic subunit. Notably, the LxxIxE binding pocket is fully exposed under these conditions suggesting that it does not contribute to the binding of Sgo1 to B56gamma. Unexpectedly, pull down assays revealed that Sgo1 seems to compete with bona fide LxxIxE containing proteins such as Kif4A and BubR1 for B56 binding. This finding was further supported by B56alpha mutants. Both, the B56 5A as well as the R222E mutant showed reduced Sgo1 binding. Surprisingly, RNAi rescue experiments revealed that the 5A, but not the R222E mutant, supports cohesion protection despite its reduced capacity to bind Sgo1. To support this finding in a complementary manner, Ueki et al introduced mutations in Sgo1. Both, Sgo1 4A and 3A mutants showed reduced/no binding to B56 and PP2A-C. LacI/LacO-based experiments confirmed reduced recruitment of PP2A-C in the case of the 3A and 4A mutants. In cells, Sgo1 4A mutant, but not the 3A mutant, rescued cohesion protection. Even in the presence of reduced B56 levels, the 4A mutant was capable of rescuing mitotic timing.

This ms describes an interesting finding which actually raises more new questions than answering old ones. This reviewer thinks that the manuscript should be published once the authors address the following points.

We thank the reviewer for these positive comments.

The Sgo1 4A mutant - despite its significant binding capacity to B56 - rescues cohesion protection. Based on this observations, the authors speculate in the discussion that either low levels of Sgo1 binding to PP2A-B56 is sufficient or binding of PP2A-B56 to Sgo1 is not required for cohesion protection. Since partial depletion of B56 did not affect mitotic timing of the 4A mutant compared to WT Sgo1, the authors argue that binding of PP2-B56 to Sgo1 might not be required for cohesion protection. To make this statement the authors have to repeat the 4A pulldown in cells partially depleted of B56 in order to confirm that under these conditions less B56 binds to the 4A mutant.

Our response: We thank the reviewer for these comments. We have performed this experiment which revealed a further reduction in B56

binding (see attached data that is not included in the revised manuscript).

However, these experiments have been challenging due to the weak binding of B56 to hSgo1 4A making it difficult to accurately quantify a further reduction by LiCor. However, we want to point out that in the revised manuscript we only note that B56 RNAi in hSgo1 4A was unable to cause a mitotic defect and interpret this as still sufficient complex formation existing. We have also rewritten the discussion (also in light of comments from reviewer 3) and emphasized more clearly that we favor a single Sgo1-PP2A-B56 pathway and that Sgo1 4A maintains sufficient binding for function. A main conclusion from our work is that the conserved pocket of B56 is more critical for Sgo1 binding than the reported interface in the structure.

Fig. 3 D and F: According to the immunofluorescence in 3D, the 4A mutant recruits much more PP2A-C than the 3A mutant. According to 3F, the difference is minor. Could the authors please address this issue. Also, the authors should quantify B56 localization in this setup.

Our response:

We reinvestigated the images again and they are correct. We think it is difficult for the human eye to evaluate quantitatively differences and are confident on the results. We have also performed hSgo1 RNAi and rescue experiments and analyzed centromeric B56alpha localization which is fully consistent with the Lac array results. We have included these new data in Fig 3D-E.

Fig. EV1F: This reviewer cannot see chromosome localization of WT B56alpha.

Our response:

We have added movies as supplemental material to make it clearer.

Previously, it has been shown that Cdk1 phosphorylation of Sgo1 is essential for cohesin binding and its cohesion protective function. The Sgo1 T346 non-phosphorylatable mutant binds PP2A and localizes to centromeres, but fails to bind to cohesin. Since the authors address the role of the Sgo1-PP2A-B56 axis in cohesion protection, this reviewer would have liked to see more experiments in this direction, e.g. how do the different Sgo1 and B56 mutants analyzed in this study behave with respect to cohesin binding. This is an important point.

Our response:

We thank the reviewer for suggesting this which turned out to be really interesting. We have investigated this now using IPs and MS analysis. By both approaches Sgo1 3A and Sgo1 4A binds more cohesin components. We interpret this as PP2A-B56 likely acting on T346 to regulate cohesin binding. Indeed, we see increased T346 phosphorylation in Sgo1 3A. These new data are included in Table EV2 and Figure 4C-D. The effect is clearly strongest with hSgo1 3A consistent with the strong reduction in PP2A-B56 binding.

Referee #3:

The authors performed a structure-function analysis of the interaction of hSgo1 (Sgol1) with PP2A-B56. The analysis was guided by a previously published crystal structure of PP2A-B56 bound to a fragment of hSgo1 and recent progress in our understanding of substrate recognition by PP2A-B56 pioneered by the Nilsson lab.

The authors provide in vitro and in vivo evidence that Sgol1 competes with LxxIxE-containing proteins for binding to B56. Accordingly, a mutation (R222E) in B56's LxxIxE-binding pocket reduces binding of Sgol1 and other LxxIxE-containing proteins. The R222E mutation increases the interkinetochore distance in nocodazole-arrested cells, an assay for protection of centromeric cohesion in mitosis. These data suggest that B56's LxxIxE-binding pocket is important for binding to Sgol1, which is, in turn, required for

protection of centromeric cohesion from the prophase pathway in human cells in culture. Whether Sgol1 actually has a functional LxxIxE motif has not been investigated.

The crystal structure published by Xu et al. revealed an interaction between Sgol1's CC domain and a surface involving residues of B56 and PP2A-C. The authors investigated B56's interaction with Sgol1 by creating mutation in B56 (B56-5A) and Sgol1. The latter mutant is called Sgol1-4A, while the mutant of Xu et al. in Sgol1 residues contacting PP2A-C is known as Sgol1-3A. B56-5A and Sgol1-4A reduce the B56-Sgol1 interaction in vitro and in vivo but less so than Sgol1-3A. While the Sgol1-3A mutation abolished protection of centromeric cohesion, the mutations B56-5A and Sgol1-4A have not detectable effect. These results let the authors favor the hypothesis that Sgol1 and PP2A-B56 have independent functions in the protection of centromeric cohesion from the prophase pathway.

Major points

Whereas I am reluctant to ask for additional experiments, I feel that the text of the manuscript has to be revised before considering publication.

Our response:

We thank the reviewer for all the comments. We have extensively rewritten parts of the discussion and also sections of the manuscript to emphasize that we favor a single PP2A-B56-hSgo1 pathway as supported by the bulk of the literature. The experiment suggested by reviewer 1 to deplete WAPL to see if this suppress the hSgo1 3A phenotype was also very instrumental in arguing that the only defect in hSgo1 3A is a cohesin defect (Fig 4G in revised manuscript). Note that WAPL RNAi does not suppress hSgo1 function in chromosome alignment arguing that these hSgo1 aspects are fully functional in hSgo1 3A. Based on these results on the comments from this reviewer we clearly state now that hSgo1 4A and B56 5A generates sufficient PP2A-B56-Sgo1 complex for function rather than our previous speculative thoughts on a two-pathway model. We emphasize that a clearly novel aspect of our work is the requirement of the fully conserved pocket on B56 for hSgo1 binding.

1. The finding that B56's LxxIxE-binding pocket is required for Sgol1 binding and protection of centromeric cohesion from the prophase pathway is important for researchers working on the function and regulation of shugoshins and centromeric cohesion in mitosis and meiosis. Given that there is no analysis of a potential LxxIxE motif in Sgol1, it would be nice if the authors could at least point out candidate motifs in the Sgol1 sequence. The Nilsson group is arguably in the best position to evaluate the potential of more derived motifs.

Our response: In the CC region of Sgo1 there is no LxxIxE like motifs and also LxxIxE motifs are always present in disordered regions (stated at the end of our discussion). We have made it more clear in the revised text that there appear to be no LxxIxE motifs in Sgo1 and that the CC region must bind in a different but overlapping region (see also comment from reviewer 1).

2. Admittedly, the literature on shugoshin is huge and the nomenclature of shugoshins in different organisms is somewhat confusing and inconsistent. It would be beneficial, therefore, if the authors could point out early on that there are two shugoshins in mammals: Sgo1/Sgol1 protects centromeric cohesion from the (non-proteolytic) prophase pathway in mitotic cells, while Sgo2/Sgol2 protects centromeric Rec8 from cleavage by separase in meiosis. In yeast, it is Sgo1 that protects Rec8 in meiosis. So, the statement at the end of the first paragraph is correct for yeast but incorrect for mammals. Since this work is on mammalian Sgol1, this creates confusion for the "untrained eye". In fact, the authors might consider calling "their" Sgo1 either hSgo1 or Sgol1 to clearly distinguish it from the yeast proteins, which are relevant to this work as some implication of the Xu et al. crystal structure have been tested in yeast.

Our response:

We thank the reviewer for these useful recommendations that we have incorporated in the revised manuscript. We refer now to Sgo1 as hSgo1 and we have clarified the roles of Sgo1 and Sgo2 in humans/yeast in the revised introduction. We also point out that our results only apply to human mitotic cells.

3. In the same vein, it would be desirable to more precisely state in the title and discussion what has been investigated here: a structure-function analysis of the interaction of Sgol1 with PP2A-B56 in human cells in culture and the implications thereof for protection of centromeric cohesion from the prophase pathway. Surprisingly, the prophase pathway has not been mentioned at all in the manuscript. To what extent these conclusions are applicable to other shugoshins or to protection of centromeric Rec8 in meiosis can or should be discussed. Ultimately, however, this remains to be investigated.

Our response:

We thank the reviewer for these useful comments. We have revised the title to: "A highly conserved pocket on PP2A-B56 is required for hSgo1 binding and cohesion protection during mitosis"- we found it a little difficult to incorporate all the suggestions from the reviewer in one short title and would also like to highlight an important conclusion from our work. We point out in the discussion that our results are restricted

to hSgo1 function in mitosis and might not apply to meiosis.

4. The citations at the end of the first paragraph should include Kitajima et al., 2006 (the Nature paper showing that centromeric Rec8 is protected by Sgo1-PP2A-B56 in fission yeast).

Our response:

We have included this now.

5. The authors find that the B56-5A and Sgol1-4A mutations weaken the Sgol1-PP2A-B56 interaction but do not seem to affect protection of centromeric cohesion. Based on this finding they promote the hypothesis that Sgol1 and PP2A-B56 have independent functions in protection. In my opinion, at least, the data do not justify the room given to this hypothesis. The data merely suggest that the protection mechanism tolerates a weakening of the interaction, which is consistent with the findings that very small amounts of Sgol1-PP2A in mammals or Sgo1-PP2A in yeast are sufficient for protection of centromeric cohesion. It is not uncommon that a reduction in the affinity of a protein-protein interaction has little effect on the process in the living cells where protein concentrations can be very high, especially for proteins accumulating at a particular structure (like the kinetochore or centromere). The authors used a similar argument for binding of the Sgol1 fragment to PP2A-B56 under crystallization conditions.

Our response: We fully agree with the reviewer on these comments. We have rewritten and shortened the discussion and do not discuss the hypothetical two pathway model. Rather we simply point out that the Sgo1 4A mutant retains sufficient PP2A-B56 binding for function while the conserved pocket on B56 seems very important for Sgo1 binding and cohesin protection. Our new results showing that WAPL RNAi fully suppress Sgo1 3A phenotype is also supportive of a common single pathway.

6. The last part of the discussion is a bit confusing and reveals a rather lax interpretation of the literature. The finding that the Sgol1-3A mutation affects several proteins is not surprising given the multitude of functions of shugoshin proteins at kinetochores/centromeres. Also, the comments with regards to expressing of soronin or Rec8 mutants are inaccurate. For instance, the Rec8 mutants, the corresponding kinase mutants, Sgo1 mutants, etc have been expressed at endogenous levels (replacing the wild-type genes) and to retain other function, such as providing cohesion or protein interactions. The authors risk, unnecessarily in my opinion, confusing the general reader or antagonizing researchers in the shugoshin field.

Our response:

We thank the reviewer for these comments. As pointed out above we have rewritten the discussion to avoid antagonizing researchers in the field and mainly focused on our results shedding new light on the Sgo1-PP2A-B56 complex. We also highlight the new results on PP2A-B56 bound to hSgo1 regulating cohesin binding.

Dear Jakob,

Thank you for submitting your revised manuscript. It has now been seen by all of the original referees. As you can see, the referees find that the study is significantly improved during revision (please note that referee #1 did not have additional comments, but he/she just informed us of his/her support for publication). Before I can accept the manuscript, I need you to address the additional points below:

- Please provide 3-5 keywords for your study. These will be visible in the html version of the paper and on PubMed and will help increase the discoverability of your work.
- We think that one of the funders for Guillermo Montoya in our manuscript submission system may have been entered incorrectly. Please double check.
- We note that Figure EV3A is currently not called out in the text.
- We note that EV datasets are movies. Please rename the files and their callouts in the text as Expanded View Movie 1, 2 and 3 (i.e. EV Movie 1 etc.). The movies should be zipped with their legends. Please remove the movie legends from the manuscript files.
- The nomenclature of the datasets should be Dataset EV#. Their legends should be removed from the manuscript text and added directly to the dataset file.
- Please split the source data into one file per figure.
- Please make the mass spectrometry data (PXD024532 and MassIVE MSV000087003) publicly available.
- Papers published in EMBO Reports include a 'synopsis' and 'bullet points' to further enhance discoverability. Both are displayed on the html version of the paper and are freely accessible to all readers. The synopsis includes a short standfirst summarizing the study in 1 or 2 sentences that summarize the paper and are provided by the authors and streamlined by the handling editor. I would therefore ask you to include your synopsis blurb and 3-5 bullet points listing the key experimental findings.
- In addition, please provide an image for the synopsis. This image should provide a rapid overview of the question addressed in the study but still needs to be kept fairly modest since the image size cannot exceed 550x400 pixels.
- Our production/data editors have asked you to clarify several points in the figure legends (see attached document). Please incorporate these changes in the attached word document and return it with track changes activated.

Thank you again for giving us to consider your manuscript for EMBO Reports, I look forward to your minor revision.

Kind regards,

Deniz

--

Deniz Senyilmaz Tiebe, PhD
Editor
EMBO Reports

Referee #2:

The authors have done a substantial amount of work to improve the manuscript and to support their data. I recommend publication as it is.

Referee #3:

In my previous comments, I asked for more clarity in presenting our current knowledge of the roles of mammalian shugoshins in mitosis and meiosis and a more careful interpretation of the author's results in the light of published findings. The authors have addressed all of these concerns in their current manuscript. In addition, the authors have performed additional experiments (in response to reviewers 1 and 2), which further support, in my view, their revised conclusions and interpretations. I therefore feel that this manuscript is now suitable for publication in EMBO Reports.

- Please provide 3-5 keywords for your study. These will be visible in the html version of the paper and on PubMed and will help increase the discoverability of your work.

We have included this on the first page and in the online form.

- We think that one of the funders for Guillermo Montoya in our manuscript submission system may have been entered incorrectly. Please double check.

We have corrected this.

- We note that Figure EV3A is currently not called out in the text.

Corrected – the figure is now called out.

- We note that EV datasets are movies. Please rename the files and their callouts in the text as Expanded View Movie 1, 2 and 3 (i.e. EV Movie 1 etc.). The movies should be zipped with their legends. Please remove the movie legends from the manuscript files.

Corrected

- The nomenclature of the datasets should be Dataset EV#. Their legends should be removed from the manuscript text and added directly to the dataset file.

Corrected

- Please split the source data into one file per figure.

Corrected

- Please make the mass spectrometry data (PXD024532 and MassIVE MSV000087003) publicly available.

The datasets have been released.

- Papers published in EMBO Reports include a 'synopsis' and 'bullet points' to further enhance discoverability. Both are displayed on the html version of the paper and are freely accessible to all readers. The synopsis includes a short standfirst summarizing the study in 1 or 2 sentences that summarize the paper and are provided by the authors and streamlined by the handling editor. I would therefore ask you to include your synopsis blurb and 3-5 bullet points listing the key experimental findings.

We have added synopsis and bullet points on the first page.

- In addition, please provide an image for the synopsis. This image should provide a rapid overview of the question addressed in the study but still needs to be kept fairly modest since the image size cannot exceed 550x400 pixels.

We have provided an image.

- Our production/data editors have asked you to clarify several points in the figure legends

(see attached document). Please incorporate these changes in the attached word document and return it with track changes activated.

We have made all our changes in the word documented with track changes.

Dear Jakob,

Thank you for submitting your revised manuscript. I have now looked at everything and all is fine. Therefore, I am very pleased to accept your manuscript for publication in EMBO Reports.

Congratulations on a nice work!

Kind regards,

Deniz

--

Deniz Senyilmaz Tiebe, PhD
Editor
EMBO Reports

--

At the end of this email I include important information about how to proceed. Please ensure that you take the time to read the information and complete and return the necessary forms to allow us to publish your manuscript as quickly as possible.

As part of the EMBO publication's Transparent Editorial Process, EMBO reports publishes online a Review Process File to accompany accepted manuscripts. As you are aware, this File will be published in conjunction with your paper and will include the referee reports, your point-by-point response and all pertinent correspondence relating to the manuscript.

If you do NOT want this File to be published, please inform the editorial office within 2 days, if you have not done so already, otherwise the File will be published by default [contact: emboreports@embo.org]. If you do opt out, the Review Process File link will point to the following statement: "No Review Process File is available with this article, as the authors have chosen not to make the review process public in this case."

Should you be planning a Press Release on your article, please get in contact with emboreports@wiley.com as early as possible, in order to coordinate publication and release dates.

Thank you again for your contribution to EMBO reports and congratulations on a successful publication. Please consider us again in the future for your most exciting work.

Yours sincerely,

Deniz Senyilmaz Tiebe, PhD
Editor
EMBO Reports

THINGS TO DO NOW:

You will receive proofs by e-mail approximately 2-3 weeks after all relevant files have been sent to our Production Office; you should return your corrections within 2 days of receiving the proofs.

Please inform us if there is likely to be any difficulty in reaching you at the above address at that time. Failure to meet our deadlines may result in a delay of publication, or publication without your corrections.

All further communications concerning your paper should quote reference number EMBOR-2020-52295V3 and be addressed to emboreports@wiley.com.

Should you be planning a Press Release on your article, please get in contact with emboreports@wiley.com as early as possible, in order to coordinate publication and release dates.

Corresponding Author Name: Jakob Nilsson

Manuscript Number: EMBOR-2020-52295V1